# BLACK-BOX TARGETED ADVERSARIAL ATTACK ON SEGMENT ANYTHING (SAM)

## ABSTRACT

Deep recognition models are widely vulnerable to adversarial examples, which change the model output by adding quasi-imperceptible perturbation to the image input. Recently, Segment Anything Model (SAM) has emerged to become a popular foundation model in computer vision due to its impressive generalization to unseen data and tasks. Realizing flexible attacks on SAM is beneficial for understanding the robustness of SAM in the adversarial context. To this end, this work aims to achieve a targeted adversarial attack (TAA) on SAM. Specifically, under a certain prompt, the goal is to make the predicted mask of an adversarial example resemble that of a given target image. The task of TAA on SAM has been realized in a recent arXiv work in the white-box setup by assuming access to *prompt* and *model*, which is thus less practical. To address the issue of prompt dependence, we propose a simple yet effective approach by only attacking the image encoder. Moreover, we propose a novel regularization loss to enhance the cross-model transferability by increasing the feature dominance of adversarial images over random natural images. Extensive experiments verify the effectiveness of our proposed simple techniques to conduct a successful black-box TAA on SAM.

## 1 INTRODUCTION

Deep learning has achieved remarkable progress in the past decade, with a trend shifting from task-specific learning to foundation models Bommasani et al. (2021). In the NLP field, its progress is featured by seminal foundation models like BERT Devlin et al. (2018) and GPT Brown et al. (2020); Radford et al. (2018; 2019). Built on top of GPT, ChatGPT Zhang et al. (2023b) has revolutionized our understanding of AI. Moreover, such large language models also accelerate the development of various cross-modality generative tasks, such as text-to-image Zhang et al. (2023e) and text-to-speech Zhang et al. (2023f) and text-to-3D Li et al. (2023). The progress of cross-modality foundation models in computer vision is also notable, such as CLIP Radford et al. (2021) and its improved variants Jia et al. (2021); Yuan et al. (2021). More recently, the Meta research team has released the *Segment Anything* project Kirillov et al. (2023), which has attracted significant attention in the field of computer vision. The resulting foundation model is widely known as SAM Kirillov et al. (2023), which performs prompt-guided mask prediction for segmenting the object of interest.

Prior to the advent of SAM Kirillov et al. (2023), numerous task-specific image recognition models, ranging from image-level to pixel-level label prediction, are widely known to be vulnerable to adversarial examples that change the model output by adding imperceptible perturbation to the image input Szegedy et al. (2013); Goodfellow et al. (2015); Madry et al. (2018). More intriguingly, the predicted label after such an attack can be preset to that of a target image, often termed targeted adversarial attack (TAA). Conceptually, a non-TAA is considered successful if the predicted label is different from the original predicted label with an assumption that the prediction is correct before the attack. For the TAA, it is considered successful only if the predicted label matches the preset target label, which is more challenging than the non-targeted one. It remains unknown whether the foundation model SAM is also vulnerable to adversarial examples for TTA, for which our work conducts the first yet comprehensive investigation. Specifically, we investigate whether the SAM can be fooled by adversarial attacks to make the prompt-guided mask of adversarial examples resemble that of the target images under a certain prompt.

The SAM pipeline works in two stages: generating the embedded features with an image encoder and then generating the masks with a prompt-guided mask decoder. Considering this pipeline, a straightforward way to implement TAA on SAM is to perform an end-to-end attack on the mask decoder, which is termed Attack-SAM in a recent arXiv work Zhang et al. (2023c). Attack-SAM, however, assumes that the attacker knows the prompt type and position. Such a dependence on the prior knowledge of the prompt makes Attack-SAM less practical because the SAM often works in an online mode to allow the user to set the prompt randomly. We experiment with increasing the cross-prompt transferability by applying multiple prompts in the optimization process and find that there is a significant trade-off between the performance of training prompts and test ones. To this end, we propose a simple yet effective approach by only attacking the image encoder. Due to its prompt-free nature by discarding the mask decoder, we term it prompt-agnostic TAA (PATA). The generated adversarial examples of PATA have embedded features that resemble those of the target images, which makes them partly transferable to unseen target models Ilyas et al. (2019). However, this cross-model transferability is conjectured to be limited by the strength of adversarial features indicated by their dominance over other clean images for determining the feature response. We verify this conjecture by adding a regularization loss to increase the feature dominance. To avoid dependence on other natural images, we propose to replace them with random patches cropped from the to-be-attacked clean image.

Overall, with the SAM gaining popularity in the computer vision community, there is an increasing need to understand its robustness. Prior work Zhang et al. (2023d) shows that SAM is vulnerable to adversarial attack but mainly in the white-box setup by assuming access to the *prompt* and *model*. By contrast, this work studies the adversarial robustness of SAM in a more practical yet challenging black-box setup. Our contributions are summarized as follows.

- With a focus on the transferability property, to the best of our knowledge, this work conducts the first yet comprehensive study on a TAA on SAM in a black-box setup.

- We identify the challenges in end-to-end approaches for realizing cross-prompt transferability and mitigate it in a prompt-agnostic manner by only attacking the image encoder.

- We identify relative feature strength as a factor that influences the cross-model transferability, and propose to boost them by a novel regularization loss.

## 2 RELATED WORK

### 2.1 ADVERSARIAL ATTACKS

Deep neural networks are found to be susceptible to adversarial examples Szegedy et al. (2013); Biggio et al. (2013), which can fool the model with wrong predicted results. Numerous works have then studied different techniques to attack the model, which can be classified from two aspects: the adversary's goal and the adversary's knowledge. According to the adversary's goal Xu & Yang (2020); Zhang et al. (2020; 2022), the adversarial attack can be divided into untargeted and TAAs. With an image classification task as an example, the untargeted attack fools the model to predict any wrong label that differs from the ground truth. Unlike the untargeted attack, the TAA is successful only when the predicted label is the same as a certain preset target label. From the perspective of the adversary's knowledge, the attacks can be divided into white-box attacks and black-box attacks. In the white-box setup, the adversary has access to all knowledge of the model, including the architecture, parameters, and gradients. The white-box setup is often adopted to evaluate the robustness of the model instead of a practical attack. By contrast, the adversary in the black-box setup has no access to the to-be-attacked model, including the model itself and other inputs (like prompt in the case of SAM). FGSM Goodfellow et al. (2015) and PGD Madry et al. (2018) are widely used for generating adversarial examples in the white-box setting. Multiple works Dong et al. (2018; 2019); Xie et al. (2019) are proposed to improve the attack in the black-box setting. The seminal ones include updating the gradients with momentum (termed MI-FGSM) Dong et al. (2018) boosts with momentum, smoothing the gradients with kernel (termed TI-FGSM) Dong et al. (2019), and resizing the adversarial example for input diversity (termed DI-FGSM) Xie et al. (2019). More recently, it has been shown in Zhao et al. (2021); Zhang et al. (2022) that combining the above techniques constitute simple yet strong approaches for transferable targeted attacks. In contrast to them mainly investigating image classification tasks, this work focuses on the task of attacking SAM.

## 2.2 SEGMENT ANYTHING MODEL (SAM)

Since the Meta released the Segment Anything Model (SAM), numerous papers and projects have emerged to investigate it, mainly focusing on two perspectives: its functionality and robustness. Given that SAM can segment anything and generate masks, a family of works investigates its performance in various scenarios. On the one hand, multiple works Zhang et al. (2023g); Han et al. (2023); Tang et al. (2023) experiment SAM in different fields, including medical images Zhang et al. (2023g), glass Han et al. (2023), camouflaged objects Tang et al. (2023), and semantic communication Tariq et al. (2023), where the results show that the SAM performs well in the general setting except the above challenging setups. Moreover, the capability of SAM for data annotation is evaluated in the video He et al. (2023b;a;b), which is critical for the video tasks in the computer vision, and SAM can generate high-quality mask annotation and pseudo-labels. On the other hand, multiple projects have attempted to extend the success of SAM to the different computer vision tasks, including text-to-mask (grounded SAM) IDEA-Research (2023), generating labels Chen et al. (2023), image editing Kevmo (2023), video Gaomingqi (2023); Zhang et al. (2023h), and 3D Adamdad (2023); Chen (2023). Grounded SAM IDEA-Research (2023) is the pioneering work to combine Grounding DINO and SAM, which aims to detect and segment anything with text inputs. Specifically, the Grounding DINO is based on the input text to detect anything, and the SAM is used to segment them. Given that SAM generates masks without labels, some works Chen et al. (2023); Park (2023) combined SAM with CLIP and ChatGPT for semantic segmentation of anything. Beyond the segmentation object, SAM is also used for image editing, such as Magic Copy Kevmo (2023), which focuses on extracting the foreground using the capability of segmenting anything of SAM. Beyond using the SAM to process the 2D images, the investigation of SAM in 3D and video has been explored, where Anything 3D Objects Adamdad (2023) combines the SAM and 3DFuse to reconstruct the 3D object. Another family of works Zhang et al. (2023d); Qiao et al. (2023) has investigated the robustness of SAM from different aspects. A comprehensive robustness evaluation of SAM has been conducted in Qiao et al. (2023), which reports the results from the style transfer and common corruptions to the occlusion and adversarial perturbation. In Qiao et al. (2023), the SAM performs robustness except for adversarial examples. Attack-SAM Zhang et al. (2023d) provides a comprehensive study on its adversarial robustness and shows the success of TAA. Their investigation, however, is mainly limited to the white-box scenario, while our work focuses on the more challenging black-box setup.

## 3 BACKGROUND AND TASK

### 3.1 SAM STRUCTURE AND WORK MECHANISM

As introduced in Kirillov et al. (2023), Segment Anything Model (SAM) contains three components: an image encoder, a prompt encoder, and a mask decoder. The image encoder adopted by SAM is a Vision Transformer (ViT) pretrained by MAE He et al. (2022), whose function is to embed the image feature. The other two components can be seen as prompt-guided mask decoder, as depicted in Figure 1, whose role is to map from the image embedding to a mask determined by the given prompt. The mask generated by SAM is determined by two inputs: image and prompt. Therefore, (x, $prompt^{(k)}$, $mask^{(k)}$) could be used for the data pair of SAM, where multiple prompts can be given, and k is the $k$-th data pair. The forward process of SAM is summarized as follows.

$$y^{(k)} = SAM(promt^{(k)}, x; \theta) \tag{1}$$

where $y^{(k)}$ is the confidence of being masked for each pixel of the image with given $prompt^{(k)}$, and $\theta$ represents the model parameter. The matrix $y^{(k)}$ has the same shape as the input image, where the height and width are H and W, respectively. We denote the coordinates of the pixel in the image $x$ as $i$ and $j$, and the mask area is determined by the predicted value $y_{ij}^{(k)}$. Specifically, $y_{ij}^{(k)}$ belongs to the masked object area when it is larger than the threshold of zero. Otherwise, it is considered as the background, unmasked region.

### 3.2 TARGETED ADVERSARIAL ATTACK (TAA) ON SAM

Before introducing the Targeted Adversarial Attack (TAA) on SAM, we revisit it in the classical classification task. We define $f(\cdot, \theta)$ as the target model: $\mathbb{X} \to \mathbb{Y}$, where $\mathbb{X}$ and $\mathbb{Y}$ represent the input

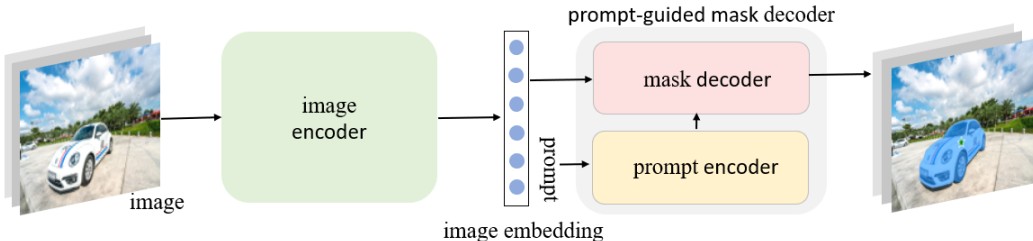

Figure 1: The process of Segment Anything Model.

images and the output labels. Let $x \in \mathbb{X}$ represent an image from the image domain $\mathbb{X}$ with the shape of H×W×C, where H, W, and C are the height, width, and channel, respectively. Let $y \in \mathbb{Y}$ denote a label from the label domain $\mathbb{Y}$, where y is the ground truth label of x. TAA aims to generate an adversarial image that fools the model to predict a target label $y_t$, which is formally stated as:

$$\delta^* = \max_{\delta \in \mathbb{S}} \mathcal{L}(f(x_{clean} + \delta; \theta), y_t), \tag{2}$$

where $\delta$ is the perturbation, and $\delta \in \mathbb{S}$ constraints the range of $\delta$ to ensure imperceptible perturbation. In traditional classification tasks, the TAA modifies the image to obtain the target labels. Except for changing the optimization target from a target label to a mask extracted from the target image, the task of TAA on SAM also differs by requiring access to the prompt. For the position type, this work mainly considers prompt point but also report the results of the prompt box. Overall, the goal of TAA on SAM is to make the predicted mask of the adversarial example resemble that of the target image under a certain prompt.

## 3.3 EVALUATION UNDER PGD-BASED ATTACK

We employ the Intersection over Union (IoU), a metric widely used in image segmentation, to quantify the success of TAA on SAM. The IoU is calculated between the adversarial and target image masks. As shown in Equation 3, the mean Intersection over Union (mIoU) computes the average IoU of N data pairs.

$$mIoU = \frac{1}{N} \sum_{i=1}^{N} IoU(Mask_{adv}, Mask_{target}). \tag{3}$$

The value of mIoU ranges from 0 to 1, with a higher value indicating a more effective TAA. Without loss of generality, we randomly select 100 images from the SA-1B dataset Kirillov et al. (2023) for mIoU evaluation. Projected Gradient Descent (PGD) Madry et al. (2018) is a widely used gradient-based attack method to generate adversarial examples. PGD is also sometimes referred to I-FGSM Kurakin et al. (2017), *i.e.* iterative variant of fast gradient sign method (FGSM) Goodfellow et al. (2015). To avoid confusion, we use the term PGD and adopt it in this work to update the perturbation gradient. Following Attack-SAM Zhang et al. (2023c), we set the step size to $l_\infty$-bounded $2/255$ with $8/255$ as the maximum limit for the perturbation.

## 4 BOOSTING CROSS-PROMPT TRANSFERABILITY

For the task of TAA on SAM, our work is not the first to solve it, Attack-SAM Zhang et al. (2023c) showed success in making the predicted mask of the adversarial image align with that obtained from a target image. To obtain the mask of the adversarial image, the attackers need to know the prompt type (point or box) and position (location on the image). An intriguing property of SAM, as shown in the official demo, is that it enables the users to apply the prompt in an interactive mode. In other words, it might not be practical to have prior knowledge of the prompt beforehand and thus it is important to make the crafted adversarial examples have cross-prompt transferability property. To this end, we first conduct a preliminary investigation to increase the transferability of Attack-SAM between different prompts. For simplicity, we assume the prompt type is point and limit the goal of cross-prompt to cross-position.

## 4.1 CROSS-PROMPT ATTACK-SAM

With the prompt type chosen as point, we aim to make the crafted adversarial examples position-agnostic. The vanilla Attack-SAM method for achieving TAA adopts a fixed position for the selected prompt. We test and find that the practice of fixing the prompt position in every iteration of the PGD attack causes over-fitting to the chosen position. To avoid over-fitting and thus making it position-agnostic, we experiment with choosing multiple random point prompts to attack the SAM. Following Attack-SAM, the optimization goal is set to

$$\delta^* = \min_{\delta \in \mathbb{S}} \sum_k ||SAM(prompt^{(k)}, x_{clean} + \delta) - Thres(SAM(prompt^{(k)}, x_{target}))||^2, \quad (4)$$

where the threshold $Thres(SAM(prompt^{(k)}, x_{target}))$ is set to a positive value if $SAM(prompt^{(k)}, x_{target}) > 0$. Otherwise, it is set to a negative value. For the value choices, we follow Atttack-SAM Zhang et al. (2023c) to set them to -10 and 40.

We randomly select 64 random positions as the test prompt points. For the number of training prompt points, we set it ranging from 1 to 100. The performance of TAA on SAM is shown in Figure 2. We can observe that the increase of training prompt points indeed helps increase transferability to unseen test prompt points. When it is set to 100, the mIoU values on the training and test prompt points are 57.57% and 47.03%, respectively. It suggests that there is a limit to further increasing the mIoU under the end-to-end attack method.

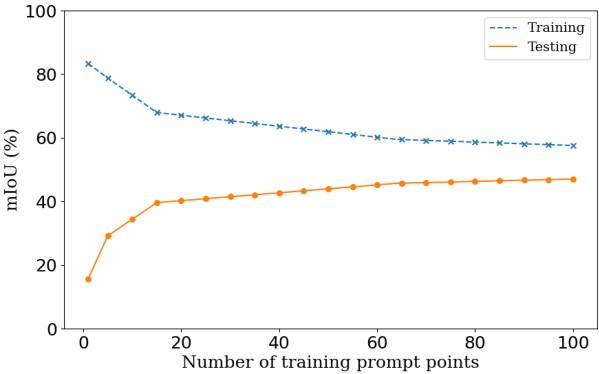

Figure 2: The mIoU results of cross-prompt Attack-SAM for training and testing with increasing training prompt points.

## 4.2 OUR PROPOSED METHOD

The above preliminary investigation of increasing the number of number $K$ of training prompts shows partial success of increasing cross-prompt transferability to unseen prompts. Theoretically, as $K$ goes to infinite, the mIoU gap between training prompts and test ones will approach zero. Therefore, we can expect that the theoretically maximum mIoU on test prompts is equivalent to the mIoU on the training prompts when $K$ is set to infinite. In practice, however, it is challenging to enumerate all prompts in the optimization process (note that considering the box prompt type can further increase the challenges). More importantly, there is a conflict between the optimization of different prompts, which causes a lower mIoU on the training prompt points as $K$ increases. Since the goal of increasing $K$ is just to make the generated adversarial examples agnostic to the prompt choice, we propose directly optimizing the perturbation in a prompt-agnostic manner by discarding the prompt-guided mask decoder. The learning goal is to make the adversarial examples have features that are similar to that of the target image without considering the prompt position type or position. In other words, $SAM_{embedding}(prompt, x_{clean} + \delta)$ is optimized to close the $SAM_{embedding}(prompt, x_{target})$. Formally, the optimization goal is presented as follows:

$$\delta^* = \min_{\delta \in \mathbb{S}} \mathcal{L}(SAM_{embedding}(prompt, x_{clean} + \delta), SAM_{embedding}(prompt, x_{target})). \quad (5)$$

Due to the prompt-free nature of Eq. 5, we term it prompt-agnostic target attack (PATA).

To make the embedded adversarial features resemble target image features, we experiment with various loss functions, including cosine, Huber, and MSE loss. The results of different loss functions are shown in Table 1. We observe that different loss functions have little effect on the mIoU results. Given that MSE loss is simpler yet outperforms other loss functions by a small margin, we adopt MSE loss for the proposed PATA in this work.

Table 1: The mIoU (%) results of three different loss functions for TAA on SAM.

| Loss functions | Cosine | Huber | MSE |
|---|---|---|---|
| mIoU | 74.52 | 75.48 | 75.71 |

**Relationship between Eq. 4 and Eq. 5.** We can interpret that the collective goal of optimizing the perturbation with $K$ prompts in Eq. 4 lies in making the embedded features similar to those of the target image. In other words, Eq. 4 and Eq. 5 have the same optimization objectives. The difference is that Eq. 5 seeks a more straightforward approach to realize the optimization objective. It is conceptually and practically more simple. Despite its simplicity, it achieves significantly better performance than the cross-prompt Attack-SAM (75.71% vs 47.03%).

**Relationship between MobileSAM and PATA.** Here, we discuss its conceptual similarity to a recent finding in MobileSAM Zhang et al. (2023a). Specifically, MobileSAM Zhang et al. (2023a) identifies the difficulty of distilling the knowledge from a heavyweight SAM to a lightweight one lies in the coupling optimization of the image encoder and mask decoder. Therefore, it only distills the heavyweight image encoder while keeping the lightweight decoder fixed. The distilled image encoder works surprisingly well with the fixed mask decoder to predict the desired mask. Conceptually, the optimization target of Eq. 5 and that in MobileSAM are the same in the sense of making the embedded student (adversarial) features similar to the embedded teacher (target) features. Their key difference lies in that MobileSAM optimizes a network through distillation, while Eq. 5 optimizes a budget-constrained perturbation on the input space.

## 5 BOOSTING CROSS-MODEL TRANSFERABILITY

In the above section, we have explored how to realize prompt-agnostic targeted attack in the white-box setup, where the attacker is assumed to have access to the model architecture and parameters. This assumption does not hold in practical scenarios when the model owner is aware of the attack risk. Therefore, we study whether it is possible to perform the above task in the black-box setup by exploiting the transferability property of adversarial examples. It is revealed in Ilyas et al. (2019) that the transferability of adversarial examples lies in the fact that adversarial examples are not bugs but features. The adversarial examples generated by PATA have embedded features that are similar to that of the target image, which therefore enables them partly transferable to unseen SAM models.

### 5.1 RELATIVE FEATURE STRENGTH

Intuitively, adversarial examples with stronger feature strength are likely to transfer better from the surrogate model to the target model. In this work, we follow Zhang et al. (2020) to measure the relative strength of adversarial features by measuring their dominance over other clean images in terms of the feature response in a competitive framework. Specifically, we denote the features of the adversarial image and those of a random competition image as $f_{adv}$ and $f_{com}$, respectively. With Cosine Similarity denoted as $CosSim$, the *feature dominance* ($fd$) is defined as

$$fd = CosSim(f_{adv}, f_{mix}) - CosSim(f_{com}, f_{mix}), \quad (6)$$

where $f_{mix}$ denotes the features of the mixed image, a sum of the adversarial and competition image. Intuitively, the mixed feature response $f_{mix}$ is determined by both the adversarial and competition

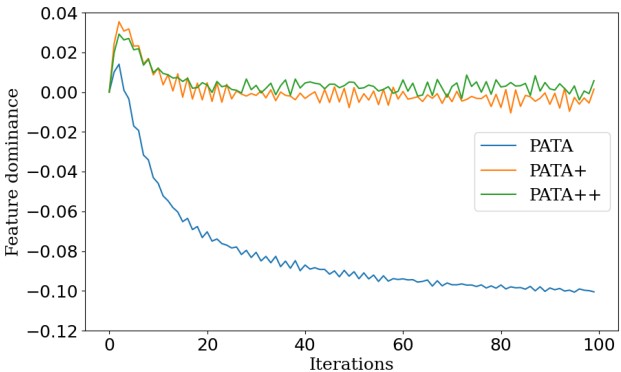

Figure 3: Feature dominance of PATA, PATA+, and PATA++.

images. In essence, $fd$ depicts the dominance of the adversarial image for determining the final $f_{mix}$. A higher $fd$ indicates a higher (relative) strength of adversarial examples in the competitive framework. In practice, to reduce randomness caused by the choice of competition images, we randomly select $M$ competition images and report the average value of $fd$ over them. When the iteration step of adversarial attack is zero, the adversarial image is the same as the original to-be-attacked clean image and therefore statistically $fd$ is expected to be around zero. Interestingly, we find that the value of $fd$ gets lower as PATA optimizes the adversarial features toward the target images with more iterations of attack optimization, which is depicted in Figure 3. An adversarial example with less dominant features is more fragile and thus difficult to transfer.

For making a successful TAA, the adversarial examples need to have adversarial features that sufficiently resemble the target image. This requires more attack iterations, which in turn decreases its relative strength in terms of feature dominance over random clean images. Motivated by this rationale, on top of the basic MSE loss function, we experiment with PATA+ by using the same number of iterations but adding a straightforward regularization loss to improve the relative strength of adversarial features. The results in Figure 3 show that PATA+ powered with the proposed regularization loss significantly increases the strength of adversarial features and thus enhances the cross-model transferability. However, this straightforward approach has some drawbacks, which will be addressed in the following.

**Practical Implementation of the Regularization.** The above regularization method has two underlying drawbacks: (a) requiring $N$ times more computation resources than the normal PATA; (b) requiring access to $N$ times more clean images (as the competition images) in the optimization process. To mitigate drawback (a), we propose to use a single competition image but change it in every single iteration, which reduces the computation overhead. For the dependence on other clean images in drawback (b), we propose to use patches randomly cropped from the to-be-attacked clean image as the competition images. We term our final attack approach with this new regularization method PATA++, which requires no external clean images and is computationally efficient. The final loss is shown as follows:

$$Loss_{total} = Loss_{mse} + \lambda Loss_{reg}, \tag{7}$$

where $\lambda$ indicates the weight of the regularization loss $Loss_{reg}$.

## 5.2 CROSS-MODEL RESULTS

As stated in Section 3, the SAM consists of an image encoder and a prompt-guided image decoder. SAM has three variants based on the different image encoder architectures: SAM-B with the image encoder ViT-B, SAM-L with the image encoder ViT-L, and SAM-H with the image encoder ViT-H. The three variants have the same decoder structure (but with different parameters). To not lose generality, we use SAM-B as the surrogate (white-box) model in the whole paper, and here we report its results on SAM-H and SAM-L. Table 2 shows the results of the cross-model attacks. It shows that the proposed PATA++ achieves a higher IoU than PATA by a large margin.

Table 2: The mIoU (%) results of all black-box models by using the ViT-B as source model.

| Attack Method | ViT-L | ViT-H |
|---------------|-------|-------|
| PATA | 26.36 | 22.78 |
| PATA+ | 29.46 | 25.91 |
| PATA++ | 30.92 | 26.58 |

**Qualitative results.** We visualize the results of the cross-model attack based on the prompt point and box in Figure 4 and 5, respectively. We randomly select the prompt point and box to show the predicted masks of adversarial examples. it shows the results of PATA++ realizing TAAs on different variants of SAM. The results of the segment everything mode are shown in Appendix (see Figure 6).

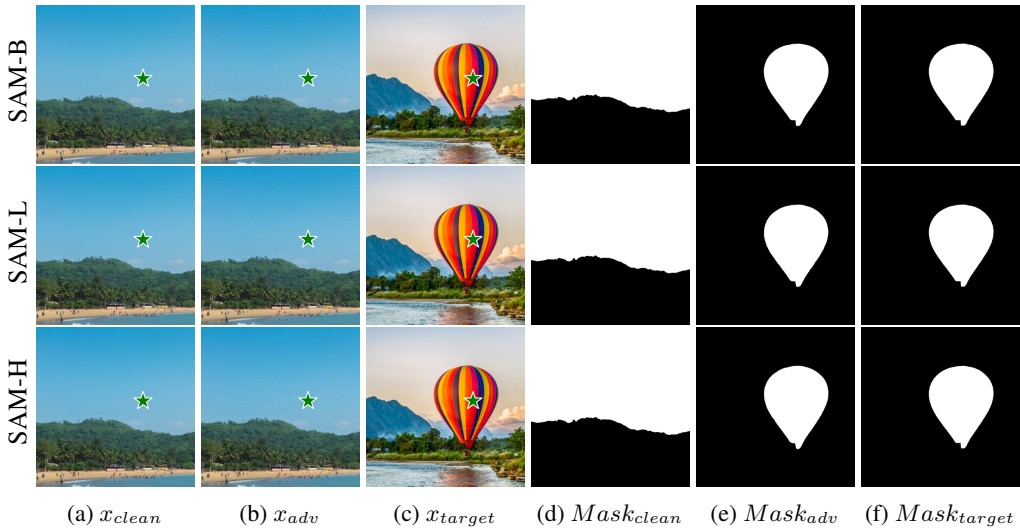

(a) $x_{clean}$    (b) $x_{adv}$    (c) $x_{target}$    (d) $Mask_{clean}$    (e) $Mask_{adv}$    (f) $Mask_{target}$

Figure 4: Cross-model results of PATA++ under point prompts.

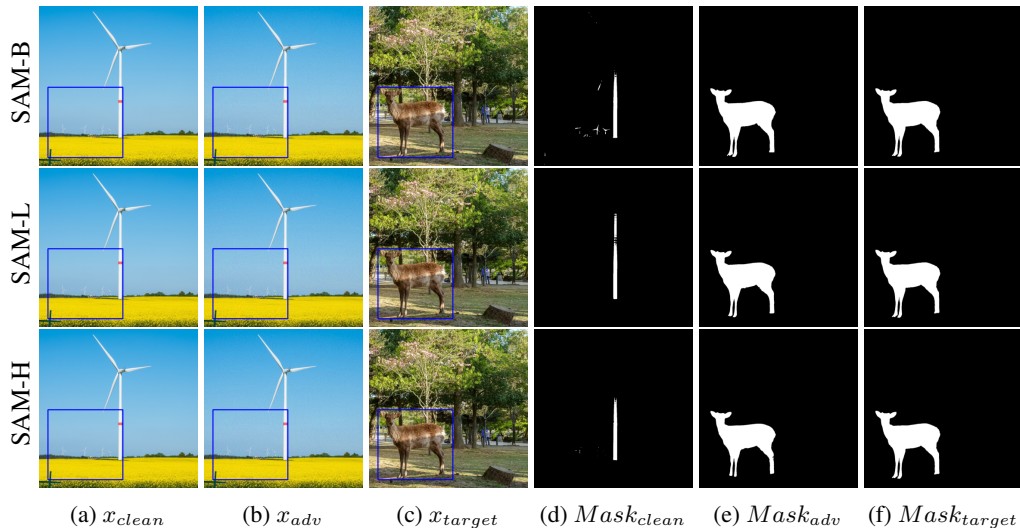

(a) $x_{clean}$    (b) $x_{adv}$    (c) $x_{target}$    (d) $Mask_{clean}$    (e) $Mask_{adv}$    (f) $Mask_{target}$

Figure 5: Cross-model results of PATA++ under box prompts.

**Ablation study.** The ablation study on the regularization weight is shown in Table 3. We set $\lambda$ to 0.01 in this work. We also experiment with different values of perturbation budgets. Following Kurakin

et al. (2018), we adopt the maximum perturbation values $\epsilon \in \{4/255, 8/255, 12/255, 16/255\}$, with results reported in Table 4.

Table 3: Effect of regularization parameter $\lambda$. We report the results with the black-box attack setting with results reported in the form of ViT-L/ViT-H.

| $\lambda$ | 0 | 0.001 | 0.01 | 0.1 |
|---|---|---|---|---|
| mIoU | 26.36/22.78 | 28.94/25.26 | 30.92/26.58 | 14.24/13.33 |

Table 4: Effect of perturbation budget. We report the results with different values of the maximum perturbation $\epsilon$ under the black-box setting with results reported in the form of ViT-L/ViT-H.

| $\epsilon$ | 4/255 | 8/255 | 12/255 | 16/255 |
|---|---|---|---|---|
| mIoU | 22.03/20.07 | 30.92/26.58 | 38.14/31.57 | 41.73/35.17 |

**Relationship with SOTA techniques.** Cross-model TAA has long been perceived as a challenging task. As discussed in Section 2, recent works Zhao et al. (2021); Zhang et al. (2022) have revealed that combining MI-FGSM Dong et al. (2018), TI-FGSM Dong et al. (2019) and DI-FGSM Xie et al. (2019) is sufficient for achieving satisfactory performance for TAA. Conceptually, the proposed regularization loss is similar to the above three techniques in the sense that they all regularize the input gradients. Therefore, we experiment with them in the context of PATA, if applicable, and discuss their effects. The results are shown in Table 5. First of all, it is challenging to apply DI-FGSM since resizing the adversarial example changes the size of the feature maps, causing difficulty in applying the MSE loss. Moreover, changing the adversarial sample size also requires adaptation of the position embedding, which further increases the difficulty of applying DI-FGSM. For the MI-FGSM, somewhat surprisingly, we find that it actually decreases the cross-model transferability. We leave a comprehensive investigation of this phenomenon to future works. Instead, we conducted a preliminary investigation and found that MI-FGSM yields a lower value of $fd$, which might explain why MI-FGSM decreases the cross-model transferability. For the TI-FGSM, we find that it improves our PATA by a visible margin, less significant than our proposed regularization loss. Combining our regularization loss with TI-FGSM achieves the highest cross-model transferability.

Table 5: Comparison with SOTA methods. Our PATA++ is complementary with SOTA techniques for further improving the TAA performance.

| Attack Method | ViT-L | ViT-H |
|---|---|---|
| PATA-MI | 22.77 | 20.97 |
| PATA-TI | 27.10 | 25.18 |
| PATA++ | 30.92 | 26.58 |
| PATA++-MI | 30.51 | 26.18 |
| PATA++-TI | 33.53 | 30.16 |

## 6 CONCLUSION

This work conducts the first yet comprehensive study on TAA on SAM in a black-box setup, assuming no access to *prompt* and *model*. Experimenting with improving the cross-prompt transferability of the end-to-end attack method by increasing the number of training prompts, we find that their collective goal is to make the embedded features of the adversarial image similar to that of the target image. Our proposed PATA realizes this goal in a conceptually simple yet practical manner. Given that PATA generates target image features, we find that its cross-model transferability can be limited by the relative feature strength. Our proposed regularization to increase the feature dominance enhances the cross-model transferability by a non-trivial margin. Extensive quantitative and qualitative results verify the effectiveness of our proposed method PATA++ in black-box setup to attack the SAM. The success of our attack method raises an awareness of the need to develop an adversarially robust SAM.

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

## A  APPENDIX

Figure 6 shows the cross-model attack results of PATA++ at *segment everything* mode. We observe that the $Mask_{adv}$ well resemble $Mask_{adv}$ for both surrogate models and unseen target models.

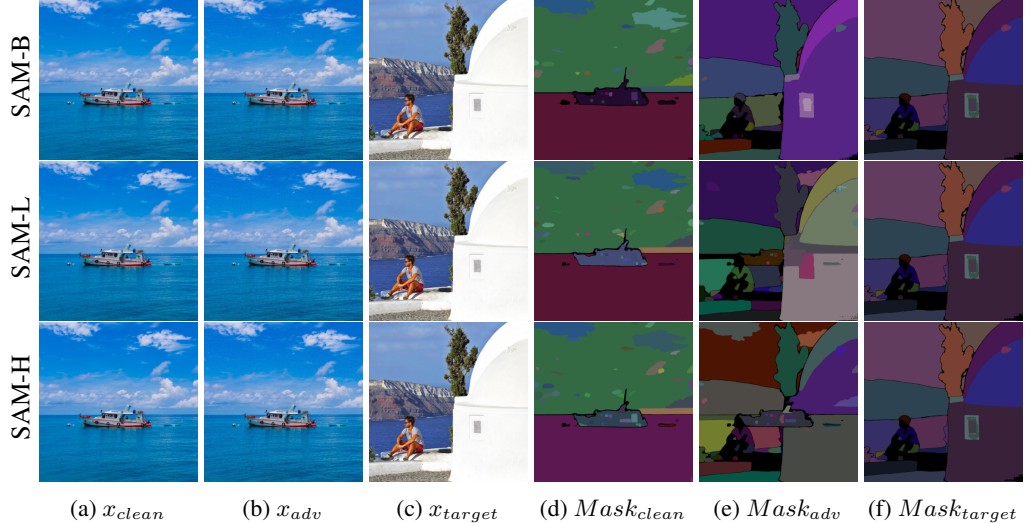

(a) $x_{clean}$  (b) $x_{adv}$  (c) $x_{target}$  (d) $Mask_{clean}$  (e) $Mask_{adv}$  (f) $Mask_{target}$

Figure 6: Cross-model results of PATA++ at *segment everything* mode.

