# OpenReview forum: "Black-box Targeted Adversarial Attack on Segment Anything (SAM)"
_ICLR.cc/2024/Conference — ICLR 2024 Conference Withdrawn Submission_

### Official Review · Reviewer_ifzZ · 2023-10-29

**Soundness:** 2 fair
**Presentation:** 2 fair
**Contribution:** 2 fair
**Rating:** 3
**Confidence:** 4

**Summary:**

this work studies the adversarial robustness of SAM in a more practical black-box setup. This cross-model transferability is conjectured to be limited by the strength of adversarial features indicated by their dominance over other clean images for determining the feature response. It verifies this conjecture by adding a regularization loss to increase the feature dominance. To avoid dependence on other natural images, it proposes to replace them with random patches cropped from the to-be-attacked clean image.

**Strengths:**

It identifies the challenges in end-to-end approaches for realizing cross-prompt transferability and mitigate it in a prompt-agnostic manner by only attacking the image encoder.

It identifies relative feature strength as a factor that influences the cross-model transferability, and propose to boost them by a regularization loss.

**Weaknesses:**

Although it claims to be black-box attack, the techniques are mainly white-box attack method and transfer attack increasing the transferability of adversarial examples. Besides, in the white-box, it uses SAM-B as the surrogate model to attack. In the black box setting, it uses the adversarial examples generated with SAM-B to attack other models with very similar architectures. It is better to transfer from an architecture to another different architecture, such as from vgg to resnet in related previous papers. Transferring with the same model architecture family makes the problem much easier.

It only tests with 100 images and report the results on the 100 images. 100 images seem to be a very small number and there may be some randomness in the results. It is better to discuss why 100 images is a reasonable number.

It is better to discuss the complexity of the proposed method, like computation complexity or what are the costs to run the algorithm.

It is hard to understand equation (5). The specific definition of $SAM_{embedding}(prompt, x_{target})$ is not provided. Equation (5) is claimed to be prompt-free. However, it still has prompts as the inputs for $SAM_{embedding}$. So it seems that equation (5) depends on prompts.

There are no baselines. Since the paper mainly uses white-box attack methods and transfer the adversarial examples. It is better to use some white-box attacks to generate and transfer the adversarial examples as a baseline. Without a baseline, we are not sure if the mIoU results are good or not. It is better to provide some basic baselines.

The technique contribution may be limited. In general, it seems that the mse loss is basically requiring the model outputs of adversarial examples and target images to be the same. This objective and the optimization is very basic in adversarial attacks. The regularization loss mainly follows Zhang et al. (2020). The novelty may be limited.

**Questions:**

see the weakness.

**Details Of Ethics Concerns:**

As the paper proposes practical blackbox attacks for large models SAM, it is better to discuss the potential social effects in the paper.

---

### Official Review · Reviewer_bXHA · 2023-11-01

**Soundness:** 2 fair
**Presentation:** 2 fair
**Contribution:** 2 fair
**Rating:** 5
**Confidence:** 4

**Summary:**

The paper proposes a targeted adversarial attack with black-box setting on Segment Anything Model (SAM), where given an adversarial image, the SAM will constantly output a predefined mask. The paper conjectures that white-box access to SAM models is impractical, where SAM works in online modes and allows users to specify random prompts. Conversely, the paper relaxes the white-box access assumption and utilizes a surrogate SAM for black-box adversarial optimizations, and ensures the attack’s effectiveness in a prompt-agnostic manner. The paper proposes a regularization loss term to further improve the adversarial attack effectiveness and computational efficiency.

**Strengths:**

- Relaxed the assumption of white-box access for adversarial attacks on SAM, where the paper proposes to use a surrogate SAM and optimize adversarial examples from it
- Improved the threat model in adversarial attacks on SAM by introducing a prompt-agnostic adversarial attack. Previous method focuses on attacking SAM with a predefined prompt position, whereas with the proposed method, the users are susceptible to a wider range of adversarial attacks, even with different prompt positions.
- Analyzes the relative feature strength between the clean image and adversarial image to identify the feature dominance between adversarial features and clean features. The paper provides a clear analysis and explanation on the impact of adversarial features and clean features, and gain insights to improve the strength of adversarial features by introducing a regularization term.

**Weaknesses:**

- The claim of black-box might be too strong, as in usual adversarial attacks, black-box settings require validation on the transferability of adversarial attacks onto different families of model. The current setting in this paper is restricted to only ViT image encoder in SAM, which is not a proper evaluation of black-box adversarial attacks’ transferability.
- The formulation of Loss_reg is unclear in Eqn. 7, and the difference between the Loss_reg for PATA+ and PATA++ is not shown.
- Cross-model results are not convincing, as the surrogate model is essentially coming from the same genre of models as the evaluated victim model. All models have similar encoder and decoder structures, but with different scales and number of parameters.
- Insufficient qualitative results to show the prompt-agnostic behavior of the proposed adversarial attack. There is no similar visualizations as Fig. 4 to demonstrate the proposed adversarial attack’s prompt-agnostic behavior.
- Typo in Eqn. 1: promt → prompt

**Questions:**

Please see weaknesses

---

### Official Review · Reviewer_Rvz4 · 2023-11-02

**Soundness:** 2 fair
**Presentation:** 2 fair
**Contribution:** 2 fair
**Rating:** 1
**Confidence:** 5

**Summary:**

This paper proposes a Targeted Adversarial Attack (TAA) on the Segment Anything Model (SAM) called Prompt-Agnostic TAA (PATA). Different from prior works, this paper doesn't require the prior knowledge of user's prompt to the SAM for choosing the object and replace with the target object.

**Strengths:**

- Adversarial attack on SAM is interesting.

**Weaknesses:**

- I do not believe Eq. 2 has to be interpreted as a classification task. If you are focussing on the image encoder, you can just compare the features from the encoder. This Eq. 2 seems unnecessary.

- Fig. 1 should accompanied by the example how TAA works with SAM, e.g. visualizing $Mask_{target}$ and $Mask_{adv}$ within the attack framework.

- Using MSE loss over other mentioned loss functions is a known phenomenon [A, B]. Table 1 is unnecessary.

- Eq. 6 is similar to loss functions used in generative adversarial attacks, where information from "other clean images" are used to create perturbations. The behavior of Eq. 6 is expected in Fig. 3.

- The paper severely lacks in comparison with prior dense prediction attacks. SAM doesn't ensure the proposed method is incomparable with these prior works.

[A] Learning transferable adversarial perturbations, NeurIPS 2021

[B] GAMA: Generative Adversarial Multi-Object Scene Attacks, NeurIPS 2022

**Questions:**

None.

---

### Official Review · Reviewer_JQeG · 2023-11-03

**Soundness:** 2 fair
**Presentation:** 3 good
**Contribution:** 2 fair
**Rating:** 6
**Confidence:** 4

**Summary:**

This paper presents a novel approach to targeted adversarial attacks (TAA) on the Segment Anything Model (SAM), a computer vision model. The contributions of the paper are noteworthy and add to the existing body of knowledge in this field. The introduction of "prompt-agnostic target attack" (PATA) is designed to generate adversarial examples that resemble a given target image's mask without relying on specific prompts. This approach has the potential to enhance the robustness and practicality of adversarial attacks on SAM, making it more applicable to real-world scenarios. The paper's emphasis on relative feature strength is another valuable contribution. This concept offers a new perspective on assessing the dominance of adversarial features over clean images, which is crucial for understanding the transferability of adversarial examples. It adds depth to the evaluation of adversarial attacks and their impact on computer vision models. Furthermore, the regularization method proposed to boost feature strength is a practical and effective addition. The authors show that this regularization significantly enhances the transferability of adversarial features across different models. This has implications for improving the overall effectiveness of TAA methods and their ability to fool various models. In summary, this paper provides novel insights and approaches for TAA on SAM, enhancing the field of adversarial attacks in computer vision. The contributions made in this paper can lead to improved robustness and security in computer vision models, particularly those like SAM.

**Strengths:**

The paper is well-structured, written clearly, and logically presented. The introduction of "prompt-agnostic target attack" (PATA) and the concept of "relative feature strength" are innovative perspectives in adversarial attacks.

**Weaknesses:**

(1)	The paper does not provide a comprehensive comparison or benchmarking against existing methods. A more in-depth evaluation against other state-of-the-art adversarial attack techniques would help assess the uniqueness and effectiveness of the proposed approach. For example, I recommend the results of adding Attack-SAM methods in the section 5.
(2)	The author mentioned in the qualitative results that the prompt information was randomly selected, but the result in the figure 4 and 5 only gives the result in the case of the prompt information to a significant area. This is not rigorous, and this ignores the important position of the prompt-guided to attack. I suggest adding experiments under the condition of different prompt-guided selection conditions.

**Questions:**

(1)	I think that if the prompt-guided is ignored, the process of generating confrontation disturbances will transfer the highly concerned area in the original picture to the regional of strong attention in the target picture. Therefore, when the prompt information is characterized by a more obvious area, it will have a good attack effect, which does not prove the effectiveness of the method. I think the results of comparative experiments to prove the effectiveness of the method.
(2)	Do you verify the concealment of the attack? From Figure 4, we can clearly see that the watermarks are similar to the outline of the target picture.
(3)	What is the direct regularization loss on Page 7? You should use formulas to explain to facilitate readers to understand.
(4)	Could you explicitly outline the limitations of the proposed approach? For instance, are there specific scenarios or model architectures where PATA may not be as effective? Addressing potential vulnerabilities or shortcomings would enhance the paper's transparency.